# APE: A Post-Training Enhancement Framework for Time Series Anomaly Detection based on Agentic Pseudo-Anomaly Generation

Zijie Chen     Shuqi Zhao

Tsinghua University

{chenzj, zhaosq}@tsinghua.edu.cn

April 29, 2026

**Proposal Due Date:** April 28th, 11:59PM
**Presentation Due Date:** April 28th, 8:00PM

**Team Members:** Zijie Chen, Shuqi Zhao
**Date:** 2026.04.28

## Contents

# 1 Introduction

Time Series Anomaly Detection (TSAD) is a fundamental capability required for the robust operation of modern complex systems, ranging from industrial manufacturing to IT infrastructure. Because anomaly labels are notoriously scarce in real-world environments, unsupervised reconstruction models have become the dominant paradigm. However, a major bottleneck in these standard approaches—such as VAEs and deep Autoencoders—is their tendency toward over-generalization. High-capacity models often successfully reconstruct both normal and anomalous patterns, causing normal and anomalous samples to become deeply entangled and difficult to separate in the representation space. This project introduces APE, a model-agnostic Post-Training Enhancement Framework based on Agentic Pseudo-Anomaly Generation (PAG), to shift the TSAD paradigm from normal-only reconstruction to anomaly-aware enhancement.

## 1.1 Gaps Identified

Current approaches that attempt to solve the representation entanglement often rely on heuristic Pseudo-Anomaly Generation (PAG). However, these methods exhibit significant shortcomings. First, they suffer from severe distribution misalignment. Fixed anomaly templates (such as manually adding spikes, noise, or cutoff operations) rely on hand-crafted rules that often deviate entirely from the actual real-world anomaly mechanisms present in specific scenarios. Second, existing PAG methods are characterized by rigid backbone coupling. Current anomaly-guided training mechanisms are typically integrated directly into a specific model's architecture, preventing them from being utilized as versatile enhancements for the broader landscape of existing TSAD backbones.

## 1.2 Novelty / Contribution

This project proposes a unified, model-agnostic framework that decouples pseudo-anomaly generation from the underlying TSAD backbone. The core novelty is the **Agentic-PAG** module, which operates in a closed loop to autonomously mine structured anomaly priors (mechanisms, morphologies, contextual priors, and scales) from multimodal scenario inputs, including raw time series, images, and domain text, utilizing advanced Vision-Language Models like Qwen3-VL. Furthermore, we introduce a **Three-Branch Contrastive Learning** post-training strategy. By effectively filtering, augmenting, and sampling these high-fidelity pseudo-anomalies, our framework reshapes the representation space, forcing normal and anomalous patterns apart via an InfoNCE-based objective without requiring changes to the original backbone.

# 2 Motivation

Unsupervised reconstruction models remain highly appealing because they theoretically assume training data is almost entirely normal and aim to learn a normal

manifold exclusively from normal patterns. The premise is that an anomaly will naturally yield a massive reconstruction error. However, the reality of deploying high-capacity deep learning models contradicts this assumption. Because these models are optimized strictly for reconstruction rather than discrimination, they learn to reproduce whatever sequence is fed into them, essentially overriding the reconstruction error metric.

We are motivated by the pressing need to transition from "normal-only reconstruction" to "anomaly-aware enhancement." While generating pseudo-anomalies is a logical step, using rigid, heuristic templates creates a distribution mismatch where the model learns to identify artificial artifacts rather than real contextual deviations. By leveraging recent advancements in multimodal AI agents to mine scenario-specific priors and converting them into an executable PAG Rule Bank, we can provide TSAD models with dynamic, high-fidelity negative samples. This allows us to explicitly pull normal representations together and push anomalous ones apart, fundamentally solving the over-generalization bottleneck.

# 3   Background

Time Series Anomaly Detection (TSAD) involves identifying unexpected, rare deviations within continuous sequential data. The standard framework historically utilizes VAE-based (Variational Autoencoder) architectures. In these setups, an input time series $X$ is fed into an Encoder $q_{\varphi}(z|x)$ to generate parameters $\mu$ and $\sigma$ for a latent representation $z$. A Decoder $p_{\theta}(x|z)$ then attempts to reconstruct the input as $\hat{X}$. The network is optimized over a dataset $D_{normal}$ to minimize the Negative Log-Likelihood and the Frobenius norm reconstruction error.

Pseudo-Anomaly Generation (PAG) introduces the concept of creating artificial anomalies to provide the model with a boundary condition. Instead of just minimizing reconstruction error, the model is trained to differentiate between the true normal data and the injected pseudo-anomalies via representation space separation techniques.

# 4   Related Works / Existing Methods

## 4.1   Unsupervised Reconstruction Models

The backbone of current TSAD relies heavily on unsupervised architectures. Classic models like VAEs (e.g., FCVAE, OmniAnomaly) focus on probabilistic latent space modeling, while more recent architectures leverage LSTMs for sequential memory (e.g., LSTM-AE) or Transformers for attention-driven global contexts. While highly adept at mapping the normal manifold, all these models inherently lack discriminative boundaries because they are never exposed to negative samples during standard training.

## 4.2 Heuristic Pseudo-Anomaly Generation

To inject abnormal knowledge, researchers have developed heuristic PAG methods, such as CutAddPaste. These methods apply fixed signal processing techniques—like FFT/IFFT manipulation for frequency anomalies, adding random noise, cutting off segments, or generating sudden amplitude spikes. While computationally cheap, these hand-crafted perturbation operators fail to reflect the complexity of domain-specific anomalies, leading to the distribution mismatch where the generated pseudo-anomalies only cover a fraction of the real anomaly space.

## 4.3 Pseudo-Anomaly Guided Training

Recent state-of-the-art frameworks like PAMA, NCAD, and PCRTA attempt to integrate pseudo-anomalies directly into the training loop via contrastive learning or binary classification. For instance, PAMA uses dual-memory augmentation to assist in contrastive learning. However, these methods tightly couple the pseudo-anomaly generator with the specific detector architecture, preventing their use as plug-and-play enhancements for the broader ecosystem of existing normal-only backbones.

# 5 Challenges

## 5.1 Distribution Misalignment in Anomaly Generation

The most fundamental challenge is ensuring that generated pseudo-anomalies share the same distribution mechanics as real, unseen anomalies. Because different domains (e.g., data center servers vs. weather sensors) exhibit drastically different anomaly morphologies, a one-size-fits-all heuristic operator inevitably generates unrealistic artifacts.

## 5.2 Backbone Coupling and Generalization

Designing a post-training framework that can operate on top of *any* pre-trained model (from FCVAE to Transformers) is mathematically complex. The enhancement phase must reshape the latent representation space without destroying the underlying normal manifold that the specific backbone has already learned.

## 5.3 Evaluation Environment Limitations

Public datasets like the UCR archive are vital but lack explicitly controlled anomaly semantics. To rigorously validate whether an Agentic-PAG accurately captures anomaly mechanisms, we need a highly controllable synthetic dataset with diverse generation mechanisms and pixel-perfect labels, which currently does not exist.

# 6 Objectives

The primary objective is to implement and validate the APE framework as a model-agnostic, post-training enhancement tool capable of improving normal-anomaly separability across a wide spectrum of existing TSAD backbones (e.g., Donut, TranAD, TimesNet, CrossAD).

A concurrent objective is to develop the **Agentic-PAG** module, ensuring it can successfully translate multimodal scenario inputs into executable, domain-grounded PAG rules. Finally, the project aims to construct **TimeAnomaly-Bench**, a diverse, controllable testbed containing explicit anomaly semantics and morphologies, to rigorously evaluate the pipeline against normal-only baselines and heuristic SOTA like CutAddPaste and PAMA.

# 7 Proposed Methodology

## 7.1 Overall Architecture

The APE framework is logically divided into three sequential phases: Agentic-PAG (A), TSAD Backbone Training (B), and PA-Guided Post-Training (C). This modularity ensures that the anomaly generation logic is strictly decoupled from the specific detection model.

## 7.2 Step 1: Agentic-PAG (Anomaly Prior Mining)

We construct a multimodal time series analysis toolkit. The input consists of Raw Time Series, TS Images (visualized plots), and Domain Text (system logs or scenario descriptions). Utilizing a Vision-Language Model agent (Qwen3-VL) paired with standard TS-Tools, the system extracts four core elements: Anomaly Mechanism, Morphology, Contextual Prior, and Scale/Duration Prior. These elements are compiled into a scenario-specific PAG Rule Bank.

## 7.3 Step 2: Unsupervised Backbone Pre-Training

The raw TS input is passed to an arbitrary, unmodified unsupervised TSAD backbone (e.g., VAE or Transformer). During this phase, the Encoder projects the input into a latent representation space $z$, and the Decoder reconstructs it. The model learns the normal manifold by exclusively minimizing the reconstruction error ($\mathcal{L}_{\mathrm{Rec}}$) and Negative Log-Likelihood ($\mathcal{L}_{\mathrm{NLL}}$) on the normal dataset $D_{\mathrm{normal}}$.

## 7.4 Step 3: PA-Guided Post-Training

Once the detector is trained, we freeze/adjust parameters and initiate the enhancement phase. Using Context Profiling and Normal Candidate Filtering, we select raw sequences ($X_{\mathrm{raw}}$). The PAG Sampler applies the customized Rule Bank to generate high-fidelity pseudo-anomalies $X_{\mathrm{pa}} = G(X; \tau, \varphi)$. Simultaneously, an Augmentation module generates minor contextual variants $X_{\mathrm{aug}} = A(X; \psi)$.

## 7.5 Step 4: Three-Branch Contrastive Learning

The sequences $X_{\mathrm{raw}}$, $X_{\mathrm{aug}}$, and $X_{\mathrm{pa}}$ are passed through the backbone's encoder to obtain representations $z_{\mathrm{raw}}$, $z_{\mathrm{aug}}$, and $z_{\mathrm{pa}}$. We reshape the representation space using an InfoNCE loss function:

$$\mathcal{L}_{\mathrm{con}} = \mathrm{InfoNCE}(q = z_{\mathrm{raw}}, k^+ = z_{\mathrm{aug}}, k^- = z_{\mathrm{pa}}) \tag{1}$$

This explicitly pulls the representations of raw and augmented normal data together, while pushing the pseudo-anomaly representations apart. The final post-training objective combines reconstruction and contrastive loss: $\mathcal{L}_{\mathrm{post}} = \mathcal{L}_{\mathrm{rec}} + \lambda\mathcal{L}_{\mathrm{con}}$.

## 8 Dataset

The project relies on two distinct data components. First, we utilize the public UCR Time Series Anomaly Archive, which serves as a standard benchmark for unsupervised methods.

Second, because public datasets often lack explicit contextual metadata, we are developing **TimeAnomalyBench**. This is a domain-grounded synthetic dataset engineered to provide a diverse and controllable testbed. It generates sequences with varying anomaly semantics, precise generation mechanisms, and highly accurate morphological labels, enabling rigorous validation of whether the Agentic-PAG accurately mimics real-world faults.

## 9 Project Schedule

Table 1: Project Execution Plan (Weeks 11–15)

| Wk | Dates | Tasks | Details | Lead |
|---|---|---|---|---|
| 11 | Mar 25 - Mar 31 | TimeAnomalyBench Design | Grounded synthesis, public dataset supplementation. | Zijie |
| 12 | Apr 1 - Apr 7 | Agentic-PAG Setup | Multimodal toolkit, Qwen3-VL integration. | Shuqi |
| 13 | Apr 8 - Apr 14 | PAG Rule Structuring | Define anomaly priors and executable rule schemas. | Zijie |
| 14 | Apr 15 - Apr 21 | Post-Training Impl. | Code three-branch contrastive loss, parameter freezing. | Shuqi |
| 15 | Apr 22 - Apr 28 | Evaluation & Report | Run backbones, compute PR-AUC/F1, draft slides. | Both |

# 10 Progress So Far

**Completed:** We have finalized the theoretical architecture of the APE framework, specifically outlining the transition from standard normal-only reconstruction to anomaly-aware enhancement. The literature review has been thoroughly conducted, identifying key heuristic baselines (CutAddPaste, PAMA, NCAD) and target evaluation backbones spanning from classic VAEs to modern state-of-the-art models like CrossAD and TimesNet. The mathematical formulation for the three-branch contrastive learning objective (InfoNCE) has been fully derived.

**Ongoing:** We are currently focusing on the active development of TimeAnomalyBench to ensure we have a robust, controllable synthetic testbed. Simultaneously, we are integrating the Qwen3-VL model into our multimodal TS-Tools pipeline to begin initial experiments on Anomaly Prior Mining. The next immediate step is to execute the planned ablation studies to isolate the impact of the multimodal prior mining versus simple heuristic rules.

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
