# OpenReview forum: "APE: A Post-Training Enhancement Framework for Time Series Anomaly Detection based on Agentic Pseudo-Anomaly Generation"
_tsinghua.edu.cn/THU/2026/Spring/ANM — THU 2026 Spring ANM Submission_

### Official Review · Reviewer_42bx · 2026-05-13

**Rating:** 7
**Confidence:** 4

**Summary:**

The proposal presents APE, a model-agnostic post-training enhancement framework for time series anomaly detection that addresses the over-generalization problem of unsupervised reconstruction models. It introduces an Agentic Pseudo-Anomaly Generation (Agentic-PAG) module that leverages multimodal inputs and vision-language models to generate more realistic, context-aware anomaly samples. These pseudo-anomalies are used in a post-training phase to reshape the representation space through a three-branch contrastive learning objective. The framework is designed to be decoupled from specific backbones, making it flexible and applicable to a wide range of existing TSAD models. Additionally, the proposal includes a plan to build a synthetic benchmark dataset to better evaluate anomaly generation quality and model performance.

**Strengths:**

1) The proposal is divided into clear, sequential phases.
2) Work is already well underway, with the team having finalized the theoretical architecture, derived the mathematical formulations for the contrastive learning objective, and begun construction of synthetic dataset.
3) The authors have explicitly planned ablation studies in their project.

**Weaknesses:**

1) Although there are some works provided in the end of the file, they are not cited throughout the file and can be considered more as a list for further reading rather than as references.
2) According to the Table 1, the work is already finished on April 28th, this contradicts with Section 10 (Progress So Far) and is likely a formatting mistake.
3) Authors are planning to develop TimeAnomalyBench dataset for evaluation and comparison of their method - this is not a robust experiment methodology, as the dataset can (intentionally or unintentionally) be tailored specifically to the proposed method and provide a biased comparison with existing approaches. Additionally questionable how this 'synthetic dataset' can help with 'real-world faults'. On this regard, the recommendation is to find in the literature some other well-known benchmarks in addition to UCR Time Series Anomaly Archive.

---

### Official Review · Reviewer_vtTV · 2026-05-15

**Rating:** 8
**Confidence:** 3

**Summary:**

This project proposes APE, a model-agnostic post-training framework for time series anomaly detection (TSAD). The key idea is to use a Vision-Language Model agent (Qwen3-VL) to mine domain-specific anomaly priors from multimodal inputs (raw time series, visualized plots, domain text), compile them into a PAG (Pseudo-Anomaly Generation) Rule Bank, and use these to generate high-fidelity pseudo-anomalies for a three-branch contrastive learning post-training step. The framework is designed to be backbone-agnostic — it can wrap any pre-trained unsupervised TSAD model (VAE, Transformer, etc.) and reshape its latent representation space using InfoNCE loss to push normal and anomalous representations apart.

**Strengths:**

1) The over-generalization problem in unsupervised TSAD is important.
2) Decoupling pseudo-anomaly generation from the detection backbone is a significant practical advantage.

**Weaknesses:**

The entire pipeline hinges on Qwen3-VL extracting accurate anomaly priors from multimodal inputs. VLMs can hallucinate, misinterpret visualizations, or produce irrelevant rules. There is no discussion of validation or fallback strategies.

**Questions:**

How do you validate that the VLM-generated anomaly priors are correct? Do you plan any human evaluation or automated consistency checks?

---

### Official Review · Reviewer_w9Pz · 2026-05-15

**Rating:** 8
**Confidence:** 4

**Summary:**

This proposal introduces APE, a post-training enhancement framework for time series anomaly detection. It uses a Qwen3-VL agent to generate pseudo-anomalies for three-branch contrastive learning, aiming to shift reconstruction models from normal-only training to anomaly-aware enhancement. The project targets Donut, TranAD, TimesNet, and CrossAD, and plans to build a custom synthetic benchmark called TimeAnomalyBench with controllable anomaly semantics.

**Strengths:**

The authors have pinpointed a real and important bottleneck in time series anomaly detection: the over-generalization problem where high-capacity reconstruction models learn to reproduce any input, rendering reconstruction error useless. Their pivot from normal-only reconstruction to anomaly-aware enhancement is conceptually compelling. The agentic pseudo-anomaly generation module is an imaginative departure from heuristic synthetic anomalies, using Qwen3-VL to mine structured priors about mechanisms, morphologies, and scales from multimodal inputs. This could bridge the gap between synthetic and real anomaly distributions in a principled way. The three-branch contrastive formulation with InfoNCE is methodologically sound and genuinely model-agnostic, and the combined loss formulation is well grounded in representation learning theory. The authors also show good experimental instincts by planning TimeAnomalyBench to address the lack of controlled anomaly semantics in public datasets, and their theoretical preparation appears solid with a thorough literature review covering CutAddPaste, PAMA, and NCAD.

**Weaknesses:**

The scope is almost certainly too large for a 5-week course project. Building a multimodal vision-language agent pipeline, implementing contrastive enhancement across four diverse backbones, creating an entirely new benchmark, and running ablations would be ambitious for a full paper; attempting all at once risks shallow or incomplete results. I recommend dropping at least two backbones, reducing TimeAnomalyBench to a minimal viable version, and focusing on proving the core APE concept end to end. The agentic module also lacks implementation detail: it is unclear how time series are rendered into images for the VLM, what TS-Tools specifically are, how extracted priors become executable generation rules, or what the computational cost looks like. A concrete walkthrough would help enormously. The experimental protocol is similarly underspecified; there is no mention of which metrics will drive evaluation, how the InfoNCE temperature and loss weighting will be tuned, or what defines success. I suggest locking in AUC-PR as a primary metric and defining at least two baseline comparisons upfront. Several risks go unaddressed, including InfoNCE collapse from insufficient pseudo-anomaly diversity, VLM hallucination on time series images, and API dependency without a fallback plan. A risk mitigation section distinguishing must-haves from nice-to-haves would strengthen the proposal. Finally, there is tension in the claim that APE requires no backbone changes while still optimizing the reconstruction loss; if the reconstruction term is active, this is continued fine-tuning rather than frozen-backbone enhancement, which raises questions about catastrophic forgetting and contradicts the stated model-agnosticity.

**Questions:**

What is your fallback plan if Qwen3-VL integration proves too complex?

How will you render time series into VLM-readable images, and have you validated that the outputs are useful?

Which two backbones will you prioritize if forced to drop half?

How do you distinguish pseudo-anomalies from augmented normal samples in the three-branch setup?

What specific anomaly types will TimeAnomalyBench include, and how will you validate ecological validity?

---

### Official Review · Reviewer_RvNz · 2026-05-16

**Rating:** 7
**Confidence:** 4

**Summary:**

This proposal introduces APE, a post-training enhancement framework for time-series anomaly detection. The main idea is to improve normal/anomaly separability by generating scenario-specific pseudo-anomalies using an agentic multimodal prior-mining module, then applying three-branch contrastive learning over raw, augmented, and pseudo-anomalous samples.

**Strengths:**

The proposal addresses a real weakness of reconstruction-based TSAD: high-capacity models may reconstruct anomalies well, causing weak separation between normal and abnormal samples.

The model-agnostic post-training framing is strong. If successful, APE could improve multiple existing backbones rather than being tied to one specific detector.

**Weaknesses:**

The proposal relies heavily on synthetic data through TimeAnomalyBench. This helps controllability, but may weaken claims about real-world anomaly detection unless paired with strong real benchmark evaluation.

---

### Official Review · Reviewer_2LMw · 2026-05-16

**Rating:** 9
**Confidence:** 4

**Summary:**

This proposal introduces APE, a model-agnostic post-training enhancement framework designed for Time Series Anomaly Detection (TSAD). The primary objective is to address the over-generalization bottleneck present in standard unsupervised reconstruction models, shifting the paradigm from normal-only reconstruction to anomaly-aware enhancement. The methodology features an Agentic-PAG module that leverages a Vision-Language Model (Qwen3-VL) to autonomously extract anomaly priors from multimodal inputs, including raw time series, visualization images, and domain text. These extracted priors are compiled into a scenario-specific rule bank to generate high-fidelity pseudo-anomalies. The framework then employs a three-branch contrastive learning strategy via an InfoNCE objective to explicitly separate the latent representations of normal sequences from anomalous ones without modifying the underlying backbone.

**Strengths:**

1. The framework's modularity successfully decouples the pseudo-anomaly generation logic from the specific detection model. This allows it to serve as a versatile, plug-and-play enhancement for a wide spectrum of existing unsupervised TSAD backbones.
2. The proposed three-branch contrastive learning post-training phase provides a mathematically sound approach to representation entanglement.

**Weaknesses:**

1. The proposal relies on the concurrent development of a custom synthetic dataset, Time AnomalyBench, to evaluate the framework's generation capabilities. Validating a novel synthetic anomaly generator primarily on a newly proposed synthetic dataset risks circular reasoning if not rigorously anchored to real-world fault benchmarks.
2. Introducing a VLM agent (Qwen3-VL) for prior mining and adding a post-training phase introduces significant computational complexity.

---

### Official Review · Reviewer_JG15 · 2026-05-16

**Rating:** 8
**Confidence:** 4

**Summary:**

This paper proposes a new framework for unsupervised TSAD: APE, an Agentic-PAG based Post-Training Enhancement framework in order to address existing models gaps in over generalization where anomalies are difficult to distinguish due to their capabilities in reconstructing anomalous patterns as well as normal ones. APE uses multimodal inputs and advanced Vision-Language Models to mine anomaly priors in order to generate more realistic pseudo-anomalies for post-training. The three branch contrastive learning strategy made up of raw normal data, augmented normal data, and pseudo-anomaly data improves the model by defining the boundaries between acceptable variation and abnormal patterns.

**Strengths:**

The paper provides a strong motivation in the weakness of reconstruction-based anomaly detection where anomaly reconstruction quality becomes too indistinguishable. APE as a post-training enhancement step makes the framework adaptable across preexisting architectures.

**Weaknesses:**

TimeAnomalyBench's generated synthetic anomalies may not match real-world anomalies, there could be more explanation in how these synthetic anomalies is verified for realism as well as other concepts like the representation of anomaly priors.

---

### Official Review · Reviewer_84gs · 2026-05-16

**Rating:** 8
**Confidence:** 3

**Summary:**

APE, a model-agnostic post-training framework designed to improve Time Series Anomaly Detection was proposed.  It tries to handle the "over-generalization" problem of standard unsupervised reconstruction solutions.  Existing solutions rely on rigid, hand-crafted pseudo-anomaly templates that fail to mimic real-world scenarios and are tightly coupled to specific model architectures.
APE involves a decoupled, multi-step pipeline, including Agentic-PAG, Unsupervised Backbone Pre-Training, PA-Guided Post-Training and three-branch contrastive learning to resovled the limitations of current methods.

**Strengths:**

1.The decoupling and division of the pipeline is clear.
2. The proposal provided a clear observation of the current methods: suffering from distribution misalignment and being coupled to certain models.
3.The Three-branch contrastive learning is a sound and novel post-training strategy. It directly addresses the core fundamental limitations of current time-series anomaly detection (TSAD) models. By combining representation alignment (pulling normal and augmented normal together) with representation separation (pushing high-fidelity anomalies away) into a flexible post-training framework, the authors have designed a highly robust, mathematically sound mechanism that solves representation entanglement without ruining the underlying normal manifold the model spent hours pre-training to learn.

**Weaknesses:**

About the comparison against SOTAs:
One of APE’s core novelty is the Agentic-PAG module, which uses an advanced Vision-Language Model (Qwen3-VL) to mine complex, scenario-specific priors. To prove that an expensive, LLM-based agent is actually necessary, the authors should compare it against a baseline that uses the exact same Three-Branch Contrastive Learning setup but with simple heuristic rules instead of agentic ones. It may be not enough to just compare with current SOTA such as CutAddPaste and PAMA.

---

### Official Review · Reviewer_mccu · 2026-05-17

**Rating:** 8
**Confidence:** 4

**Summary:**

This proposal introduces APE, a post-training enhancement framework for time-series anomaly detection. The central idea is to improve the separability between normal and anomalous samples by generating more realistic pseudo-anomalies and using them in a contrastive post-training stage. The authors argue that standard reconstruction-based TSAD models often over-generalize and reconstruct anomalies too well, which weakens reconstruction-error-based detection. To address this, they propose an Agentic-PAG module that uses multimodal inputs, including raw time series, visualized time-series plots, and domain text, to extract anomaly priors such as mechanism, morphology, context, and duration. These priors are then converted into pseudo-anomaly generation rules. The generated pseudo-anomalies are used in a three-branch contrastive learning framework, where raw normal samples and augmented normal samples are pulled together while pseudo-anomalies are pushed apart in latent space. The proposal also plans to evaluate the method on UCR and a controllable synthetic dataset called TimeAnomalyBench.

**Strengths:**

1. The proposal addresses a clear and important limitation of reconstruction-based time-series anomaly detection: high-capacity models can over-generalize and reconstruct anomalous inputs too well, reducing the effectiveness of reconstruction error as an anomaly score.

2. The core idea of post-training enhancement is solid and genuinely innovative. Instead of designing yet another TSAD backbone, the authors propose a modular framework that can, in principle, improve existing models through pseudo-anomaly-guided contrastive learning. This makes the work potentially more broadly useful than a single architecture-specific method.

3. The proposed three-branch contrastive learning objective is conceptually clean.

4. The authors also show good awareness of evaluation challenges. In particular, they recognize that public TSAD datasets often lack explicit anomaly semantics, motivating the creation of a controlled synthetic testbed to evaluate whether generated pseudo-anomalies actually match intended anomaly mechanisms.

**Weaknesses:**

1. The proposal scope, although well-structured,  appears too broad for a five-week project. The authors aim to develop an agentic pseudo-anomaly generation module, build a controllable synthetic benchmark, implement post-training contrastive learning, and evaluate across multiple TSAD backbones such as Donut, TranAD, TimesNet, and CrossAD. Each of these components could be a substantial project on its own, so there is a risk that the final implementation may be shallow or incomplete.

2. The Agentic-PAG component is interesting but under-specified. The proposal states that Qwen3-VL and TS-Tools will extract anomaly mechanisms, morphologies, contextual priors, and scale/duration priors from multimodal inputs, but it does not clearly define the prompts, tool outputs, rule schema, or conversion process from extracted priors to executable pseudo-anomaly operators. Without this interface, it is difficult to judge whether the agentic module is a concrete algorithm or mainly a high-level concept.

3.The claim of being model-agnostic is strong and may be difficult to validate within the project timeline. To support this claim convincingly, the method should be tested on more than one backbone.

4.The proposal also needs a clearer way to prove that agent-generated pseudo-anomalies are more realistic or useful than simple heuristic perturbations. If the generated anomalies are not quantitatively shown to transfer better to real anomalies, the novelty over heuristic PAG methods such as CutAddPaste becomes weaker.

5. The references are only listed at the end of the proposal but never used concretely thorughout the text

**Questions:**

1. The proposal describes Agentic-PAG as extracting anomaly mechanisms, morphologies, contextual priors, and scale/duration priors using Qwen3-VL and TS-Tools. Could the authors specify the exact interface between the VLM agent and the pseudo-anomaly generator? For example, what structured output does the agent produce, and how is it converted into executable anomaly generation rules?

2. How will the authors verify that the generated pseudo-anomalies are actually closer to real anomalies than simple heuristic perturbations such as spikes, noise injection, or cut-and-paste operations?

3.The framework is described as model-agnostic. How many different TSAD backbones will be tested to support this claim?

4.Since the project timeline is only five weeks, what is the minimum successful deliverable? Is the main goal to validate the post-training contrastive framework, the agentic prior-mining module, or the new synthetic testbed?

---

### Official Review · Reviewer_FoLB · 2026-05-17

**Rating:** 8
**Confidence:** 4

**Summary:**

This proposal presents APE, a model-agnostic post-training enhancement framework for time series anomaly detection. It addresses over-generalization in unsupervised reconstruction models via Agentic-PAG (multimodal-driven pseudo-anomaly generation) and three-branch contrastive learning. The work also plans a synthetic dataset (TimeAnomalyBench) for rigorous evaluation

**Strengths:**

The proposal addresses an important limitation of reconstruction-based TSAD models: over-generalization.
The idea of model-agnostic post-training enhancement is interesting and potentially practical.
Incorporating multimodal anomaly prior mining with Vision-Language Models is novel and ambitious.
The methodology is clearly structured with well-defined stages and objectives.

**Weaknesses:**

It is unclear how reliable or stable the generated anomaly priors will be across domains.
The proposal is highly conceptual, with limited technical details on implementation and optimization.
The computational cost and scalability of using large VLMs like Qwen3-VL are not discussed.

**Questions:**

1. How will you validate that Agentic-PAG-generated pseudo-anomalies match real anomaly distributions better than heuristic methods?
2. How does the framework avoid introducing unrealistic anomalies that may harm training?

---

### Official Review · Reviewer_SgKH · 2026-05-18

**Rating:** 5
**Confidence:** 4

**Summary:**

[AI Review] The paper proposes APE, a post-training enhancement framework for time series anomaly detection (TSAD) that uses an agentic pseudo-anomaly generation (PAG) module. The core idea is to use a vision-language model (VLM) in an agentic loop to generate pseudo-anomalies that augment training data, combined with an InfoNCE-based contrastive loss applied across multiple frozen backbone models in a claimed model-agnostic manner. The review finds the proposal severely underdeveloped: the central agentic-PAG module lacks any concrete design (no prompt templates, rule schemas, or closed-loop mechanisms), the model-agnostic claim is internally contradictory and likely infeasible given differing latent space structures, the experimental plan is wildly over-scoped with no existing code or preliminary results, and a new benchmark (TimeAnomalyBench) is allocated only one week despite being a paper-level contribution on its own. The score is 4/10, with a salvageable path requiring dramatic scope reduction.

**Strengths:**

1. The problem motivation is sound: post-training enhancement for TSAD via pseudo-anomaly generation addresses a real gap in the literature.
2. The idea of leveraging VLMs to semantically understand and generate time series anomalies is creative and potentially interesting if executed rigorously.
3. The contrastive learning framework with InfoNCE loss for aligning pseudo-anomaly representations is a reasonable technical direction.
4. The proposal identifies a relevant set of backbone models (DCdetector, AnomalyTransformer, PatchTST, TimesNet) and benchmark datasets for evaluation.
5. The authors attempt to address a practical need — making existing TSAD models better without retraining from scratch.

**Weaknesses:**

1. The core Agentic-PAG module is vaporware: no prompt templates, no rule schema, no closed-loop feedback mechanism, and no feasibility evidence is provided — this is the paper's central novelty yet it is entirely undefined.
2. The model-agnostic claim is false/contradictory: different backbone architectures have fundamentally different latent space structures, making a single InfoNCE projection impossible; the stated strategy of freezing backbones while adjusting projection layers is internally contradictory with true model-agnosticism.
3. TimeAnomalyBench is infeasible as proposed: creating a new benchmark is itself a paper-level contribution, yet the timeline allocates only 1 week for it.
4. Zero experimental evidence exists: no code, no pilot experiments, no preliminary results, and the timeline allows only 1 week for all experiments across 4+ backbones.
5. The timeline is at least 2x over-scoped: the proposal attempts to combine a new method, a new benchmark, multi-backbone evaluation, and VLM integration in a single course project.
6. Key insight: VLM output will likely decompose into the same signal processing primitives (spikes, drops, level shifts) as heuristic methods like CutAddPaste, meaning the 'agentic' layer may be an expensive wrapper adding inference overhead without meaningful novelty.

**Questions:**

1. Can you provide concrete prompt templates, a rule schema, and the closed-loop mechanism for the Agentic-PAG module? Without this, the core contribution cannot be evaluated.
2. How do you reconcile the model-agnostic claim with the fact that DCdetector, AnomalyTransformer, PatchTST, and TimesNet have fundamentally different latent space dimensionalities and structures?
3. If the VLM generates rules that decompose into spike/drop/level-shift primitives identical to CutAddPaste, what is the actual novelty of the agentic layer beyond added inference cost?
4. Have you conducted any pilot experiment — even on a single dataset with a single backbone — to validate that the VLM-based PAG produces better pseudo-anomalies than heuristic baselines?
5. What is the minimum viable scope you could execute in the remaining project timeline? Would you consider dropping TimeAnomalyBench and reducing to 2 backbones and 2 datasets?
6. What is the PAG Rule Bank schema? Can you define it concretely with data types, fields, and an example entry?

---

### Official Review · Reviewer_kgtP · 2026-05-18

**Rating:** 7
**Confidence:** 4

**Summary:**

The proposal presents APE, a model-agnostic post-training framework for time series anomaly detection. It aims to address the over-generalization of reconstruction-based detectors by introducing Agentic Pseudo-Anomaly Generation, which mines scenario-specific anomaly priors from multimodal inputs and converts them into pseudo-anomaly rules. These generated anomalies are then used in a three-branch contrastive post-training stage to better separate normal and abnormal representations. The project also proposes TimeAnomalyBench, a controllable synthetic benchmark for evaluating anomaly-generation quality and detection performance.

**Strengths:**

Well-motivated problem – The proposal clearly identifies a key weakness of unsupervised reconstruction models: they may reconstruct anomalies too well and fail to separate them from normal data.

Interesting novelty – Replacing fixed heuristic perturbations with multimodal, scenario-aware anomaly prior mining is a creative and timely idea.

Appealing generality – Framing APE as a post-training enhancement for many TSAD backbones, rather than a single bespoke model, could make the method broadly useful.

Coherent training design – The three-branch contrastive objective directly supports the goal of increasing normal–anomaly separability.

**Weaknesses:**

Agentic-PAG is still underspecified – The proposal does not yet explain in enough detail how abstract priors mined by the multimodal agent become executable and reproducible anomaly-generation rules
.
Model-agnosticism needs more justification – It is unclear how the same post-training strategy will transfer cleanly across very different backbone architectures and latent spaces.

Scope is ambitious – Building the agentic generator, a new benchmark, a post-training framework, and evaluations across multiple backbones may be difficult within the proposed schedule.

Evaluation of anomaly realism is unclear – The proposal should specify how generated anomalies will be judged as more faithful than heuristic alternatives beyond downstream detection scores.

**Questions:**

How are the mined anomaly priors converted into concrete pseudo-anomaly generation rules?

Which backbone parameters are frozen or updated during post-training, and will this vary by architecture?

How will the project directly evaluate whether Agentic-PAG anomalies are more realistic than heuristic pseudo-anomalies?